# The Impact of Essential Amino Acids on the Gut Microbiota of Broiler Chickens

**DOI:** 10.3390/microorganisms12040693

**Published:** 2024-03-29

**Authors:** Thyneice Taylor-Bowden, Sarayu Bhogoju, Collins N. Khwatenge, Samuel N. Nahashon

**Affiliations:** 1Department of Agriculture and Environmental Sciences, Tennessee State University, Nashville, TN 37209, USA; snahashon@tnstate.edu; 2College of Medicine, University of Kentucky, Lexington, KY 40506, USA; sarayubhogoju@gmail.com; 3College of Agriculture, Science and Technology, Department of Biological Sciences, Delaware State University, Dover, DE 19901, USA; ckhwatenge@desu.edu

**Keywords:** broiler chickens, amino acids, microbiota

## Abstract

The research involving the beneficial aspects of amino acids being added to poultry feed pertaining to performance, growth, feed intake, and feed conversion ratio is extensive. Yet currently the effects of amino acids on the gut microbiota aren’t fully understood nor have there been many studies executed in poultry to explain the relationship between amino acids and the gut microbiota. The overall outcome of health has been linked to bird gut health due to the functionality of gastrointestinal tract (GIT) for digestion/absorption of nutrients as well as immune response. These essential functions of the GI are greatly driven by the resident microbiota which produce metabolites such as butyrate, propionate, and acetate, providing the microbiota a suitable and thrive driven environment. Feed, age, the use of feed additives and pathogenic infections are the main factors that have an effect on the microbial community within the GIT. Changes in these factors may have potential effects on the gut microbiota in the chicken intestine which in turn may have an influence on health essentially affecting growth, feed intake, and feed conversion ratio. This review will highlight limited research studies that investigated the possible role of amino acids in the gut microbiota composition of poultry.

## 1. Introduction

All nutrients necessary for chicken’s health and ample performance are provided by a defined formulated diet that contains energy, protein, mineral supplements, specific amino acids, and vitamins. The nutrient profiles used in feed formulations for broiler chickens are typically based on economically important production outcomes which include weight gain, feed intake, feed conversion ratio, and carcass yield [1]. The management practices of broilers, environmental stressors, and immunological challenges all drive the need for essential nutrients. The nutritional needs of the intestine must be considered when attempting to achieve optimization of the nutrient dispensing to broilers raised under different sanitary environments, as the intestine may have increased nutritional requirements to maintain its cellular proliferation [2]. An essential component of efficient poultry production is optimized nutrition. Chickens’ feed account for about 70% of total cost in chicken production, poultry diets are expensive since egg and meat production require high amounts of energy and protein sources [3]. Regulation of important metabolic avenues for growth, maintenance, and immunity as well as being a component of proteins are integral roles of amino acids. They regulate gene expression and the synthesis of hormones and molecules of countless biological importance [4]. Synthetic amino acids are commonly added to feed to correct any nutrient deficiencies that may occur throughout the growth process of poultry. The main amino acids for animal nutrition are dl-methionine, l-lysine, l-threonine, and l-tryptophan which continue to be manufactured for animal feed use principally by chemical synthesis [5].

Methionine, a precursor of succinyl-CoA, homocysteine, cysteine, creatine, and carnitine, is an essential sulfur-containing amino acid [6] Methionine is the first limiting amino acid in the diet and dictates the absorption of other amino acids such as cysteine [7,8]. In broilers, methionine helps in feather development, improved growth performance, production of antibodies, and directly influences immunity responses [9,10] Methionine has a positive effect on the expression of stress-related genes, which aids in protecting cells against oxidative stress [11,12,13,14]. Methionine is supplemented during the fattening of broilers, resulting in a better performance of animals and an increased growth of breast and leg muscles [15,16,17]. Broilers fed increasing concentrations of methionine led to a decrease in abdominal fat, an increase in growth rates as well as muscle yield in the breast, and legs [18,19]. However, methionine deficiency decreases the relative weight of the lymphoid organs, resulting in reduced growth [20]. A deficient supply of methionine has been shown to affect the chemical composition of different tissues and certain aspects of breast meat quality of broilers [21].

In corn-soybean meal for poultry production growth and maintenance is achieved by lysine, the second limiting amino acid. Lysine is very important in protein synthesis and an essential amino acid necessary for poultry nutrition. In broiler chickens, lysine is a crucial amino acid which can’t be synthesized within the bird’s body and must be supplemented through dietary means. Through metabolic pathways lysine interacts with threonine which contributes to protein utilization in animal feeding [22,23,24] Lysine supports muscle tissue formation, produces antibodies, and enzymes that are essential for growth. Previous research has shown optimum dietary levels of lysine affect amino acid balance and dietary protein levels in poultry [25,26,27]. The promotion of the conversion of amino acids into protein occurs with diets high in lysine and the resulting in high carcass yield [28,29]. In other poultry species, such as the pearl grey guinea fowl, various concentrations of lysine influence their growth performance [30]. Increasing dietary lysine increased feed consumption, and weight gain in the starter period significantly; as well as the feed to gain ratio at the grower period [31]. In contrast, decreased lysine concentration in broiler chicken diets causes high mortality rates during early development, and depleted levels of lysine produce an incredible decrease in body weight; all of which may be linked to the change in expression of neuroendocrine molecules such as ghrelin, leptin, and adiponectin [32]. 

Threonine is another indispensable amino acid and is the third limiting amino acid in poultry nutrition. An essential nutritional amino acid, due to the chicken’s inability to synthesis threonine de novo, especially if the diet consists of corn and soybean meal. Protein synthesis and the catabolism of threonine create many elements essential in metabolism [33]. Amino acid balance for the nutritional need for poultry is achieved by adding L-threonine to the diet [34]. Broiler chickens have a high threonine requirement for maintenance compared to other amino acids because it is abundantly high in the intestine and the turnover rate is higher [35]. Supplementation with threonine has been shown to improve growth performance and carcass trait [22]. Deficiencies in threonine could decrease the efficiency of methionine and lysine use [36,37]. Threonine has been reported as being involved in intestinal functionality and maintenance due to it being extracted in greater proportion by the small intestine compared to the other dispensable amino acids [38,39]. The mention of threonine’s ability to influence the intestinal function in chickens from previous studies may hint to amino acids ability to change the microbial balance within the intestine. 

Along with methionine, in chickens, tryptophan is an additional essential amino acid that has an influential role in protein structure and is essential for the synthesis of serotonin and melatonin [40]. The activity of certain neurotransmitters causing an alteration in poultry behavior has been linked to tryptophan [41,42,43,44,45,46,47]. Tryptophan is also associated with feed intake, growth, tissue repair, blood pressure, and body temperature [48,49,50]. In a most recent study, tryptophan was shown to increase humoral and cellular immunity in broiler chickens, as well as improve growth performance [51]. Deficiencies in tryptophan may have an effect on behavioral responses such as poultry stress and pecking behavior due to tryptophan being a precursor for melatonin and serotonin in the diet and other precursors of tryptophan (e.g., niacin) in poultry diet can help decrease fat synthesis in the body [52]. These factors constitute tryptophan as a critical nutrient for avian nutrition. 

The mentioned amino acids have clearly shown their health benefits for chickens. The manifestation of these benefits starts within the GIT of the chickens which are inhabited by various microbes. Figure 1 illustration shows how essential amino acids methionine (1st limiting), lysine (2nd limiting) and threonine (3rd limiting) can be added to poultry feed grain to provide the necessary amino acids needed to meet the desired productivity. The findings for the benefits of these added amino acids were looking at bird performance not gut microbiota. Those amino acids are ingested by the birds and within the GIT the microbes that are housed within share and/or absorb those amino acids. The mechanisms or processes in which this takes place are unknown or poorly understood. The purpose of this review is to showcase the possible role of 1st, 2nd and 3rd essential amino acids in the balance or abundance of various microbiota within the gut of broiler chickens. 

## 2. Shaping the Gut Microbiota Population

The balance in the population of the bacteria within the gut is crucial. Its microbial communities are known to have coevolved with their hosts to such an extent that it has been suggested that the composition of host microbiota can be as unique as that of a fingerprint [53]. The abundance of the microbial community in the gut is mostly affected by feed, age, use of feed additives, and pathogenic infections [54,55]. It has been consistently reported that the lead phylum of bacteria in the gut of chicken is *Firmicutes*, and that the range of abundance of from 50% to 90% for the total bacterial population in the ceca [56,57], while Firmicutes (mainly *Lactobacillus*) generally from more than 90% of all taxa in other segments of the gut, ileum and jejunum [56,57,58,59]. Through phylogenic and statistical analysis of 16S rRNA gene sequences recovered from intestinal microbiome of both the chickens and turkeys, a global bacterial consensus was created for poultry intestinal microbiome, yet the consensus was incomplete [60]. Bhogoju et al. 2018 [61] used a metagenomic approach to compare the microbial profile of guinea fowl and broiler chickens with the findings showing total number of microbial species detected in the chicken GIT was higher than that found in the Guinea Fowl GIT and the phylum *Firmicutes* was most abundant in both avian species whereas phylum actinobacteria was most abundant in chickens than Guinea fowls. The phylum level from broiler chicken jejunum mucosa, revealed that microbial colonies were Actinobacteria, Aminicenantes, Bacteroidetes, Chloroflexi, Firmicutes, Fusobacteria, Lentisphaerae, Proteobacteria, plus others and in the ileum and jejunum included mainly: Bacilli, Bacteroidia, Betaproteobacteria, Clostridia, Gammaproteobacteria, Deltaproteobacteria, Epsilonproteobacteria, Erysipelotrichia, Negativicutes, and others [54]. Many studies have shown the associations between the balance of these microbes within the microbiota and the effects on the overall health of the host, highlighting the importance of microbial balance of the microbiota within the GIT. Also, pointing out the microbiota diversity between various types of poultry. In humans and mice, extremes in body weight gain are related to altered intestinal microbial populations. Goodrich et al., 2014 [62] showed that *Mogibacteriaceae* resides in the gut; however, these populations in humans and mice have clustered with other organisms that are associated with lower body mass index (BMI). The same is true for changes in diet of non-obese diabetic mice supplemented with cellulose, pectin, and xylan [63]. High protein diets in rats were shown to decrease *Faecalibacterium prausnitzii* in the large intestine [64,65]. Shifts in ratios of *Firmicutes* and *Bacteriodetes* have been associated with obesity in humans [66,67] and this phenomenon was similar in the cattle rumen concerning energy harvesting and correlated increases of fat [68]. Myer et al. 2015 [69] discovered that steers differing in feed efficiency rumen microbio was a component that influenced the efficiency of weight gain at the 16S level. Changes within the gut of the host possibly affect maturity, growth rate, and immune status [55].

Gut microbiota composition and activity can be rapidly shaped by different dietary nutritional levels, nutrients, and texture. In the human small intestine, the most amino acid abundant fermenting bacteria belong to *Clostridium* clusters, the *Bacillus-Lactobacillus-Strepococcus* groups, and *Proteobacteria* [70] possibly making these bacterial groups essential for protein digestion and amino acid absorption in the GIT. This symbiont relationship is beneficial to the bacteria as well as the host in which the amino acids that these bacteria provide can also be utilized by the host for protein and energy production. The undigested and unabsorbed nutrients throughout the GI tract, more so in the distal ileum and ceca may serve as potential substrates for the residing microbiota [71]. An extensive variety of bacterial metabolites can be produced from all amino acids but most specifically valine, isoleucine, tryptophan, tyrosine, phenylalanine, lysine, and cysteine [72]. 

Gut microbiome performs such a vital role in feed digestion and absorption, creating interests with the associations between gut microbiome and the host feed utilization efficiency. Groups of bacteria were identified that may possibly be associated with broiler growth performance by using microbial profiling of broiler chickens across many feeding trials [73]. A few more comprehensive analyses using NGS also revealed certain bacteria that might be associated with growth performance of broiler chickens [74,75]. There are limited reports that observe dietary supplementation with some essential amino acids may mediate gut microbiota compositions and diversity especially in broiler chickens. Research studies with diets for poultry particularly broiler chickens and laying hens that have observed any influence that amino acids have on the microbiota of poultry have been listed in Table 1. The composition and population of microbiota of chickens has a crucial role in performance and health status [76]. Poultry’s intestinal tract possesses a complex microbial community consisting primarily of bacteria and diet is a major factor that can influence the microbial population in the intestinal tract. In young broilers, threonine has been reported to affect intestinal integrity and barrier function [77]. While in laying hens, dietary crude protein (CP) reduction decreased intestinal bacterial diversity, whereas dietary threonine (Thr) supplementation to low CP diet recovered the bacteria diversity and significantly increased the abundance of potential beneficial bacteria [78]. Supplementation with tryptophan higher than the current recommended standard ileal digestible for Trp (0.22%) produced a microbial shift in the ceacum and indicated a microbial shift towards beneficial bacteria [79]. Saeed et al., 2019 [54] reported that theanine, which is found in green tea has been reported to have health benefits, theanine in broilers feed increased number of *Lactobacillus* with age and treatment within both ileum and jejunum yet, the number of *Bacteriodes* decreased with age (at 42 d) with the treatment of L-theanine in the jejunum but increased at 21 d with treatment in ileum and decreased to 42 d in the control group and at day 21, bacterial richness and diversity were higher than at day 42 where Clostridium cluster XI and Lactobacillus were found most abundant but theanine is not a 1st, 2nd or 3rd limiting amino acid. Clustering of cecal communities using principal coordinates analysis (PCoA), showed a clear separation of microbial communities based on age (P < 0.05) of birds and between low and medium/ high levels of TSAA(DL-methionine) [80]. Lastly, in another 16S metagenomics study involving lysine-restricted piglets, intestinal microbiomes were sharply altered which; might influence higher feed intake in lysine-restricted group compared to control group [81], showing lysine restriction can alter gut microbiome in pigs.

## 3. Amino Acid Utilization in Bacteria

Amino acids regulate energy and protein homeostasis [82,83] as well as supports growth and bacterial survival [84]. They are one of the most valuable nutrient sources for bacteria and can be utilized as the sole nitrogen, carbon, and energy sources [85]. Once up take of amino acids occur in bacteria, they can either be directly incorporated into bacterial cells as protein building blocks or become catabolized. The presence of amino acid is highly efficient in *Streptococcus* activities and the microorganism requires glutamic acid, histidine, methionine, cysteine, valine, leucine, tyrosine, and lysine, which cannot be produce these essential amino acids therefore depending on exogenous nitrogen sources that utilize peptide proteins from growth medium by enzymatic activity [86,87]. Lysine has been responsible for the optimization of *Streptococcus thermophilus* growth [88]. Glucose and cysteine along with 5 other amino acids is essential in synthetic medium when growing *Virbio costioclus* [89] and synthetic medium for *Halobacterium salinarium* contains ten amino acids and cytidlyic acid; valine, methionine, isoleucine, and leucine are essential for growth [90]. In mixed ruminal bacteria, studies have shown that certain amino acids or amino acid subgroups stimulate in vitro growth yields. Dai et al. 2001 [67] proposed that small intestine microbiota uses lysine in milk-fed piglets, and the catabolism of lysine in the intestinal mucosa was found to be quantitatively greater than lysine incorporated into the mucosal proteins. Microbial fermentation of sugar has aided in the industrial production of lysine, and genetic engineering uses various strains of bacteria to enhance efficiency of production has allowed lysine to be prepared from other substances [91].

The preferred amino acid substrates of colonic bacteria include lysine, arginine, glycine, leucine, valine, and isoleucine. Bacteria can sense amino acids in their environment. The least nutritional valuable amino acids are also the non-utilized and least chemoattractant amino acids to bacteria. Also, most of the amino acids that *E. coli* was attracted to are preferentially utilized during growth, with a strong correlation between the order of utilization and the chemo attractant potency. However, in this same study *B. subtilis* did not demonstrate the same behavior having a weak response to glutamate, aspartate, arginine, and lysine which suggests that attraction to and utilization of amino acids is dependent on the physiology or the environment of the organism. In most studied bacteria, the number of amino acids that attract is on average significantly larger in environmental than in intestinal bacteria [92]. Laterally in the GI tract, exogenous and alimentary proteins are hydrolyzed into peptides and amino acids by host-and bacteria derived proteases and peptidases [93,94]. These peptides and amino acids that have been released by the bacteria can be further used by the host thus highlighting the beneficial symbiotic relationship between the host and its microbiota. Cecal bacteria can catabolize uric acid to ammonia, which can be absorbed by the host and used to synthesize a few amino acids such as glutamine [95]. Consequently, gut bacteria themselves can be a source of amino acids [96]. An in vitro study has shown chicken intestinal microbiome requires simple sugars and peptides for balanced growth [97]. More simple sugars and peptides may be available in the intestine of poultry which may have selected an intestinal microbiome adapted to simple sugars and peptides [98]. This may have established a microbiome selected for simple sugars and peptides.

Understanding how bacteria utilize amino acids may shed some light on bacteria growth or lack thereof depending on the environment and available nutrient sources. Poultry diet components are crucial in impacting the intestinal microbiome due to the escape of the host digestion and absorption providing growth substrates to the intestinal bacteria [99]. It has been reported that in comparison to corn-based diets, wheat-based diets affect numerous bacteria and just a small modification in dietary cereal grain composition can potentially affect the intestinal bacteria at strain level [100,101]. Many inhibitory factors associated with the external environment, such as lack of nutrients, will slow down the growth rate of lactic acid bacteria. Thus, demonstrating that protein level and source within the diet of poultry can influence the gut microbiota. Rations composed of corn-soybean favored *Lactobacillus agilis* type R5, while high wheat middlings favored *L. agilis* tyle R1 [102]. Additionally, fermented cottonseed meal in poultry diets have been shown to increase lactobacilli population in the cecum of broiler chickens [103]. *L. reuteri* S5, a lactobacillus strain isolated from the intestines of healthy white feather broilers genome encodes peptidases and amino acid transport systems, taking in nitrogen from the outside environment [104]. A new form of genotyping maybe potentially be found in studying the biochemical characteristics of bacteria. Yet, laboratory experiments that try to display this concept may produce inconsistencies and uniformity within their results. These differences are illustrated in Figure 2.

## 4. Recent Studies That Relate to Amino Acid Influence of Intestinal Microbiota

The bulk of the research in this review concentrates on investigating the influence of amino acids on GIT of chickens. Primarily, the focus lies on broiler chickens, considering factors such as the absence of additional feed additives, adjustments in optimal feed formulations, occurrences of disease or deformities, and instances of stress. A recently published research could provide an enhanced understanding of how amino acids might affect the intestinal microbiota of chickens. For instance, a study on Qingyuan partridge chickens revealed that enhancing dietary arginine led to improved growth performance and positively influenced the population structure of gut microbiota [105]. Similarly, in yellow feathering chickens supplemented with isoleucine, bacterial 16S rDNA full-length sequencing indicated that dietary isoleucine increased the cecal abundances of the Firmicutes phylum, as well as Blautia, Lactobacillus, and unclassified Lachnospiraceae genera. Conversely, it decreased the abundance of Proteobacteria, Alistipes, and Shigella [106].

However, it’s worth noting that arginine is not among the most limiting amino acids for broiler chickens, and isoleucine typically ranks as the 4th or 5th limiting amino acid. The current review primarily focuses on research involving the 1st, 2nd, and 3rd limiting amino acids for broiler chickens. Currently, there is a lack of research on this topic concerning lysine, which is the 2nd limiting amino acid crucial for broiler chicken growth and immunity. While past and present research on gut microbiota in broiler chickens, as well as other poultry, has demonstrated similarities in the abundance of Firmicutes and Lactobacillus, which are associated with positive gut health in other animals, more investigation is necessary. It remains to be seen whether changes in the quantity of essential amino acids provided by broiler chicken diets will affect the abundance of these microbiota in the GIT.

## 5. Future Studies

Microbiota within the gut functions like an endocrine organ to regulate host health through influencing the function of the gastrointestinal tract including diet digestion, nutrient resorption, immunity hormone synthesis, and nerve conduction [107,108]. It has been reported that poultry housed according to the European legislation, stress levels were not enough to change the microbiota composition [109]. Thus, leading to look more closely at the nutritional factors that drive the gut microbiota composition. Subsequently, the positive impact of gut microbiota of broiler chickens could be a key factor in enhancing optimal health, welfare, and ultimately boosting resilience in poultry. Manipulation of broiler chicken gut bacteria may be feasible to strengthen poultry productivity but the full understanding of the variations of those microbiota and nutritional functions is lacking. Evaluating the appropriate proportions and administering amino acids in poultry diet is essential for enhancing the cadence of the poultry gut.

The full metabolic potential of broiler gut microbial is poorly understood, which gives rise to more than just metagenomic studies to understand the behavior and interactions that broiler chicken microbiota has within their complex environment. Large-scale culture-based studies are necessary to acquire mechanistic comprehension into the functions of broiler gut microbiota [110]. Alterations of growth media by totally removing or decreasing the required amount of an essential amino acid such as lysine for a gut probiotic *L. reuteri*, caused an inhibition of growth (unpublished data) The observed inhibition of growth for this gut microbiota can have a huge impact on the balance and abundance of lactic acid bacteria in the GIT of broiler chickens thus affecting the antimicrobial activities that *L. reuteri* possesses. There may be a link between the essential amino acids needed for gut microbiota growth that translate into broiler chicken health and growth. When nutritional needs of *L. reuteri* and other lactic acid bacteria are met, the essential amino acids may modulate the enhancement of the production of metabolites that are antimicrobial to pathogenic bacteria. This preliminary study showcased statements #1 and #2 from Figure 2 provided in this review as it pertains to the metabolic state of the microbiota and the essential amino acid needs during bacterial growth. There are breaks within the knowledge of how broiler chicken gut microbiota intermingles with essential amino acids and peptides. More exploration needs to be done to understand in depth the modulation of essential amino acids and broiler chicken gut microbiota. Additionally, administering probiotics or prebiotics to the diet of boiler chickens without the use of antibiotics is considered as a more natural approach to improve poultry production.

## 6. Discussion

There are three main classes as it pertains to the development of the gastrointestinal tract of farm animals. Those three classes include omnivores, carnivores, and herbivores. Absorption of nutrients from the diet and the excretion of waste products are the primary function of the GI and the ecosystem of the gut is influenced by the flow of diet nutrients and host derivative substrates [111]. The makeup of the gastrointestinal microbiota varies between animal species, between individuals within the similar species and between the body sites of the host. Profoundly the microbiota from the gastrointestinal tract interacts with their animal host which aids in the determination of the initial development, quality of life, ageing and resistance to infectious diseases [112]. In commercial poultry production feed enzymes have been used to alter the gut environment and its connected microbiota to improve poultry performance [113]. Colonization of bacteria in the gut begins instantly after the hatching period, while the intestinal microbial composition is induced by several factors such as pathological conditions, genetics, environment, age, and diet [114]. Host diet can significantly affect the gastrointestinal tract microbiome, which in turn shapes host metabolism and welfare [115]. The intestinal microbiota, which demonstrates high diversity in the species of bacteria, is preserved in a comparative balance essential to the host’s health [60]. Chickens’ gastrointestinal tracts are havens of diverse and intricate microbiota that perform a protective barrier by attaching to the epithelial walls of the enterocyte [116]. Chicken GI microbiota includes various bacterial species, and different microbial communities are found in several sections of the chicken intestinal tract, with the most dominant phyla being *Firmicutes*, *Bacteroides*, and *Proteobacteria* [117,118,119]. The factor affecting the intestinal microbiota composition are age, sex, breed, diet, and pathogens [71]. Feed’s physical form and chemical makeup affect digestibility and nutrient absorption, which impacts the chicken to gut microbial composition [120].

The composition of microbiota can be affected or influenced by nutrition as well as the function that the microorganisms are going to present for the host [121]. The gut microbiota among humans and animals has been shown to remain a very crucial influencer of health. Various health conditions and disorders have been linked to the lack thereof types and number of microbes that are housed in the GIT. In comparison to other mammals that are food animals, poultry (chicken, turkey, and duck) has a shorter GI tract and faster digesta transit, ref. [122] which gives rise to a very different intestinal microbiome. A great importance to poultry production is the effect that the intestinal microbiome has on the interaction of the host and diet. Due to the large impact that diet plays on the intestinal microbiome of poultry dietary components such as wheat, barley rye, and corn have been studied extensively. The high starch and crude protein substances that wheat provides to broiler chickens makes it a crucial source of energy. Indeed, arabinoxylans, the primary non-starch polysaccharides found in wheat, have the capability to form highly viscous solutions. This viscosity can affect the digesta within the intestines, potentially limiting the contact between nutrients and absorption sites in the intestinal mucosa. This phenomenon has been observed to influence the performance, gut morphology, intestinal microbiota, and digesta of broiler chickens [123]. Various studies have revealed that the protective potential of the native commensal microbiota may simply be disrupted by different nutritional factors, thus compromising one of the indicated ways for the commensal microbiota to maintain performance and health of broilers [124]. In the poultry industry, the achievement of meeting the amino acid requirements of fast-growing broilers has been the major concern for the farmers, nutritionists, and commercial industry. However, amino acids of undigested proteins are usually manipulated by hind-gut microbiota [125]. Yet, the components that may escape host digestion and absorption such as amino acids, may serve as elements for growth. The free formed amino acids provided in a highly digestible protein source seem to be preferential over those less digestible sources [126]. Increased dietary amino acid density has been studied to reduce atrophy of the intestinal mucosa and maintain the balance of the microbiota [127].

## 7. Concluding Remarks

In conclusion, this review undergoes the limited research available on the influence of amino acids on the gut microbiota of broiler chickens. The impact of amino acids on the host microbiota is notably influenced by the sources of nutrients, primarily protein in nature. Before delving into specific findings in chickens, it is essential to showcase similar effects observed in human microbiota, as well as in pigs and mice, owing to their precedence in research. Emerging evidence suggests that amino acids can not only affect the GI barrier function but also influence inflammation in broiler chickens, thereby significantly impacting gut microbiota and overall health.

Functional amino acids such as lysine, arginine, methionine, glutamine, and threonine are recognized for their importance in addressing gut integrity-related issues. In broiler chickens, the GI is subjected to many challenges which alter performance, animal health, welfare and livability which are influenced by these amino acids, whether beneficial or disadvantageous to birds. Therefore, preventive measures are crucial to mitigate the impacts of the changes brought by amino acids on gut health while reducing the need to use antimicrobial agents. Further studies are vital to understand, access, and discover the role amino acids play in shaping and maintaining the gut microbiota in poultry.

## Figures and Tables

**Figure 1 microorganisms-12-00693-f001:**
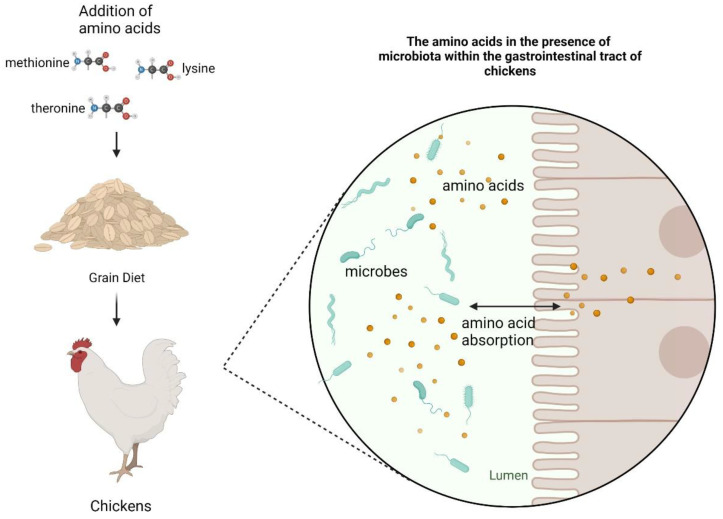
Illustration of the addition of the 1st, 2nd, and 3rd limiting amino acids to chicken feed with a snapshot of the microenvironment within the gut of the chicken. Gut microbes absorb amino acid within the environment as well as exchange or share amino acids with gut epithelia cells in the lumen of the gastrointestinal tract of the chicken. Illustration “Created with BioRender.com”.

**Figure 2 microorganisms-12-00693-f002:**
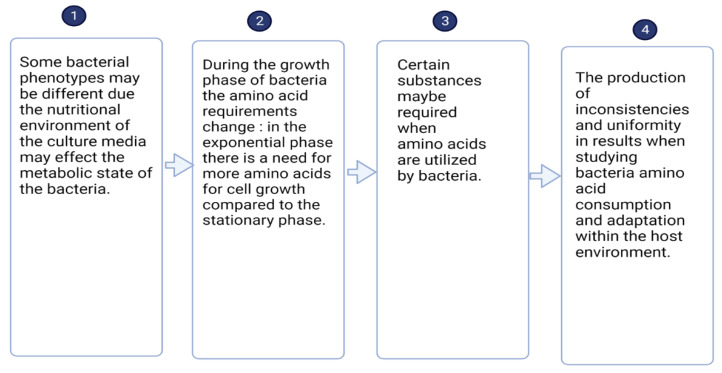
Flow diagram that details what leads to the unifromity and inconsistencies with laboratory results to attempt to showcase bacterial amino acid utilization within the host environment. Diagram “Created with BioRender.com”.

**Table 1 microorganisms-12-00693-t001:** Research studies observing amino acid effects on microbiota in broiler chickens and laying hens.

Amino Acid of Interest	Published Work	Findings
Threonine	[77,78]	Threonine supplementation fluctuates the microbial balance in the intestine. Threonine supplemented with a low CP diet brought back bacterial diversity and abundance of beneficial bacteria in laying hens.
Tryptophan	[79]	Tryptophan (0.22%) supplemented in diets of hens produced a microbial shift toward beneficial bacteria.
Theanine	[54]	Theanine supplemented in broiler chicken diets significantly increased lactobacillus compared to the control group in the ileum and jejunum.
Methionine	[80]	Clear separation of microbial communities based on age of birds and between low and medium/ high levels of TSAA (DL-methionine).

## Data Availability

All data was taken from published literature and the internet and are freely available.

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
