# Peer review of "The Impact of Essential Amino Acids on the Gut Microbiota of Broiler Chickens"

_microorganisms, 2024, doi:10.3390/microorganisms12040693_

Round 1

Reviewer 1 Report

Comments and Suggestions for Authors

The manuscript "The Influence of Amino Acids on the Microbial Flora of Broiler Chickens" by Taylo-Bowden et al. is a review of the contribution of a number of amino acids to the microflora of chickens. As many as 121 sources are cited in the review. It is not badly written and I have no particular comments to make.

I will outline a few minor points below:

Line 133: Change 16s to 16S.

Lines 180-181: Typos - remove r in threanine.

Line 185: Write what PCoA is.

Line 209: Salinarium with a lowercase letter.

Line 224: Subtilis with a lowercase letter.

Line 232: Bacteria with an lowercase letter.

Line 252: L. reuteri in italics.

Lines 265, 281 and 337: gastrointestinal tract in GI.

Line 334: Lysine, Arginine, Methionine with lowercase letter.

Author Response

Thank you very much for reviewing this manuscript. Below are the revisions as suggested. 

Line 133: Change 16s to 16S. 16S has been added to manuscript

Lines 180-181: Typos - remove r in threanine. The r has been removed. 

Line 185: Write what PCoA is. Principles of Coordinated analysis has been added to the manuscript. 

Line 209: Salinarium with a lowercase letter. salinarium with lowercase letter has been replaced. 

Line 224: Subtilis with a lowercase letter. Subtilis with lowercase letter has been replaced. 

Line 232: Bacteria with an lowercase letter. Bacteria with lowercase letter has been replaced. 

Line 252: L. reuteri in italics. L.reuteri has been italicized in the manuscript.

Lines 265, 281 and 337: gastrointestinal tract in GI. GI has replaced gastrointestinal tract in the manuscript. 

Line 334: Lysine, Arginine, Methionine with lowercase letter lysine, arginine, methionine with lowercase letters was added to the manuscript.

Reviewer 2 Report

Comments and Suggestions for Authors

Please improve table 1. Add more studies and from the studies add more informations: the type of analysies made for microbiota, the type of feed used for chickens, so one...

The conclusions should not contain citations. Move those paragraphs to discussions. In the Conclusions put your own remarks.

Adda a subchapter of Fluture perspectivei for this topic.

Author Response

Thank you very much for reviewing this manuscript. Below are the revisions as suggested. 

Please improve table 1. Add more studies and from the studies add more informations: the type of analysies made for microbiota, the type of feed used for chickens, so one...

Figure 1 Illustration shows how essential amino acids methionine (1st limiting), lysine (2nd limiting) and threonine (3rd limiting) can be added to poultry feed grain to provide the necessary proteins needed to meet the desired productivity. The findings for the benefits of  these added amino acid have been discussed in the introduction but these finding were looking at bird performance not gut flora.   The mechanisms or processes in which the amino acids are shared or absorbed by the GIT bacteria are unknown or poorly understood.   Research studies with diets for poultry that have observed any influence that of amino acids have on the microflora of poultry have been listed in Table 1. Research studies observing amino acid effects on microflora in broiler chickens.

The conclusions should not contain citations. Move those paragraphs to discussions. In the Conclusions put your own remarks.

Concluding remarks now have now citations in the text, the paragraphs have been moved to the discussions. 

Conclusions contain original remarks. 

Adda a subchapter of Fluture perspectivei for this topic

Subchapter for future perspectives for this topic have been added